# In Vivo Endophytic, Rhizospheric and Epiphytic Colonization of *Vitis vinifera* by the Plant-Growth Promoting and Antifungal Strain *Pseudomonas protegens* MP12

**DOI:** 10.3390/microorganisms9020234

**Published:** 2021-01-23

**Authors:** Marco Andreolli, Giacomo Zapparoli, Silvia Lampis, Chiara Santi, Elisa Angelini, Nadia Bertazzon

**Affiliations:** 1Department of Biotechnology, University of Verona, 37134 Verona, Italy; marco.andreolli@univr.it (M.A.); silvia.lampis@univr.it (S.L.); chiara.santi@univr.it (C.S.); 2Research Centre for Viticulture and Enology, CREA, 31015 Conegliano, Italy; elisa.angelini@crea.gov.it (E.A.); nadia.bertazzon@crea.gov.it (N.B.)

**Keywords:** bacterial inoculum, biocontrol, endophytic bacteria, epiphytic bacteria, *Pseudomonas protegens* MP12, rhizospheric bacteria, *Vitis vinifera*

## Abstract

An evaluation was conducted of the colonization of *Pseudomonas protegens* MP12, a plant-growth promoting and antagonistic strain, inoculated in vine plants during a standard process of grapevine nursery propagation. Three in vivo inoculation protocols (endophytic, rhizospheric, and epiphytic) were implemented and monitored by means of both culture-dependent and independent techniques. Endophytic treatment resulted in the colonization of the bacterium inside the vine cuttings, which spread to young leaves during the forcing period. Microscopy analysis performed on transformed dsRed-tagged *P. protegens* MP12 cells confirmed the bacterium’s ability to penetrate the inner part of the roots. However, endophytic MP12 strain was no longer detected once the plant materials had been placed in the vine nursery field. The bacterium also displayed an ability to colonize the rhizosphere and, when the plants were uprooted at the end of the vegetative season, its persistence was confirmed. Epiphytic inoculation, performed by foliar spraying of cell suspension, was effective in controlling artificially-induced *Botrytis cinerea* infection in detached leaves. The success of rhizospheric and leaf colonization in vine plants suggests potential for the future exploitation of *P. protegens* MP12 as biofertilizer and biopesticide. Further investigation is required into the stability of the bacterium’s colonization of vine plants under real-world conditions in vineyards.

## 1. Introduction

The grapevine is one of world’s commonest and most economically significant fruit crops [1]. Almost every plant organ (i.e., stem, leaves, flowers) is susceptible to attack by fungal and bacterial pathogens [2], causing several financial losses in the global wine and grape industry [3]. As a consequence, grapevine cultivation requires the extensive use of synthetic fungicides [4], but growing awareness of their dangerous side-effects on human health and the environment has led to a drive to find new alternative strategies for controlling fungal diseases.

One of the most promising of these alternative strategies is the use of bacterial strains to counter phytopathogenic fungi [5,6,7]. The antagonistic action may be exerted directly, by means of antibiosis or competition, or indirectly, by inducing plant resistance responses [8,9]. Endophytic, epiphytic, and rhizospheric microorganisms are promising biocontrol agents [10,11,12], but their reduced biocontrol effectiveness when applied in open field rather than the laboratory is hampering their adoption by agriculture. The main concern regards the colonization and persistence of the microbial inoculum in plants [13,14]. Both abiotic parameters (temperature, moisture of air and/or soil, etc.) and biotic factors (predation, antagonist, competition, etc.) can reduce persistence and therefore the antagonistic effects of microbial inoculants in the field [15]. Although there is great variation in rhizospheric, endophytic, and epiphyfitic habitats, biocontrol strains have not usually been isolated from the plant organ to which they are applied [12,16]. This may be one of the reasons for the inconsistency in their survival and effectiveness in the open field. A strain already adapted to the habitat may be a more suitable candidate. The endophytic *Burkholderia phytofirmans* PsJN can colonize the root surface, enter those roots, and then spread to grape inflorescence, stalks, and immature berries through xylem vessels, thereby protecting grapevines against *Botrytis cinerea* [17]. In vitro, the rhizospheric and plant growth promoting strain *Pseudomonas protegens* MP12 displays inhibitory effects toward different grapevine phytopathogens. Previous in vitro results had shown the ability of this strain to efficiently and permanently colonize inner grapevine roots and shoots [18], making this strain potentially useful as biocontrol agent for grapevine.

The main aim of this study was to assess the colonization and persistence of *P. protegens* MP12 in vine plants under unsterile conditions during a standard process of grapevine nursery propagation. The strain was inoculated using (i) endophytic (ii) rhizospheric and (iii) epiphytic methods and monitored by means of both culture-dependent and -independent techniques. This intention was to identify the most effective inoculation protocols for the use of *P. protegens* MP12 as a suitable and efficient biocontrol agent in the vineyard.

## 2. Materials and Methods

### 2.1. Pseudomonas protegens MP12, Botrytis cinerea BC and Plant Materials

*Pseudomonas protegens* MP12 was isolated from the rhizosphere of hardwood forests located in Brescia (North Italy). It was previously identified and characterized in vitro for both its antagonistic properties towards different phytopathogenic fungi and its ability to colonize the inner grapevine organs [18]. *Pseudomonas protegens* MP12 was grown and maintained in King B broth or agar (King B broth supplied with 1.5% *w*/*v* bacteriological agar) [19]. *Botrytis cinerea* BC was previously isolated from withered grapes [20]. The grapevine propagation process, performed in a nursery of the Vitiver group (Verona, Italy), made use of scions of Corvina cultivar and cuttings of SO_4_ rootstock.

### 2.2. Transformation of P. protegens MP12 with the Fluorescent Marker Protein dsRed

For the insertion of the fluorescent marker (dsRed protein) in *P. protegens* MP12 genome, the mini-Tn7 system was chosen [21]. Continued antibiotic selection can be avoided, because this transposon allows a site- and orientation-specific chromosomal insertion at a neutral site downstream of the *glmS* genes [21]. Chromosomal single-copy and site-specific insertion elements allows the performance of experiments in environments where antibiotic selection is not feasible without compromising the host cell fitness and performances [21]. The transformation of MP12 strain was carried out following the protocol described by Choi and Schweizer [22]. Electrocompetent *P. protegens* MP12 cells were prepared growing overnight a six mL culture in LB medium at 27 °C with shaking (225 r.p.m.). After 24 h the culture was distributed into four sterile microcentrifuge tubes and centrifuged at 16,000× *g* at room temperature for two minutes. The supernatant was removed, and each cell pellet was suspended in one mL of 300 mM sucrose. An aliquot of 100 µL of electrocompetent cells was transferred into a two mm gap-width electroporation cuvette with 50 ng of pUC18T-mini-Tn7T-Gm-dsRedExpress and 50 ng of pTNS2 plasmids (Addgene Inc, Watertown, USA). Electroporation was performed using the following settings: 25 mF, 200 O, and 2.5 kV. Immediately afterwards, 1 mL of LB medium was added, and the culture was incubated with shaking (225 r.p.m.) for 90 min at 27 °C. The culture was then plated on an LB + Gm 10 µg mL^−1^ plate and incubated at 27 °C overnight. The insertion of the dsRed marker was verified by colony PCR using glmSdown and Tn7R primers (5-TGTTAGGTGGCGGTACTTGG-3 and 5-CACAGCATAACTGGACTGATTTC-3, respectively) performed as follows: 95 °C 5′; 30 x (95 °C 45″; 59 °C 30″; 72 °C 20″); 72 °C 10′. PCR products from four transformed colonies and one from wild type *P. protegens* were cloned and sequenced to verify the correct gene insertion. Eventually, dsRed-tagged *P. protegens* MP12 was grown on King B agar medium, and the colonies were observed through Leica MZ16 F fluorescent stereo microscope. Eventually, PGPR features of the transformed strain were analyzed in vitro as described by Andreolli et al. [18].

### 2.3. Detection of Fluorescent dsRed-Tagged P. protegens MP12 Inside Root Tissue

Dormant vine cuttings were placed in water for about three weeks in order to induce root formation. The dsRed-tagged *P. protegens* MP12 was incubated at 27 °C for 48 h (200 r.p.m.), collected by centrifugation (4500× *g* for 20 min at 4 °C), and resuspended (about 10^8^ CFU mL^−1^) in half-strength sterile Hoagland’s solution [23]. Thereafter, roots free of adhering soil particles of ten one-year old plants were immersed in the bacterial suspension for one week. The same number of plants were placed in Hoagland’s solution as control. Root segments were hand sectioned and observed using confocal microscope (Nikon A1R HD25). DsRed was excited at 561 nm wavelength by solid-state lasers; emission filtering was accomplished with band-pass filters at 640/50 nm. Z-projections of root hairs were from 100 images taken at increments of 0.3 mm (MCL NanoDrive). Time-lapse images were taken at two-second increments. Stacks were processed using ImageJ software (http://rsbweb.nih.gov/ij/). Exposure times were 300 ms.

### 2.4. Plant Inoculation

#### 2.4.1. Endophytic Inoculation Treatments

*Pseudomonas protegens* MP12 was grown for 48 h at 27 °C (200 r.p.m.) until the stationary phase was reached (about 10^9^ CFU mL^−1^). Afterwards, cells were centrifuged (4500× *g* for 20 min at 4 °C) and re-suspended in half-strength sterile Hoagland’s solution [23]. The final bacterial suspension contained about 10^8^ CFU mL^−1^. Two different endophytic inoculations were performed: (i) inoculum A: 100 rootstock cuttings and 100 Corvina scions (4–6 internodes), during the hydration before grafting, were immersed in the bacterial suspension for 24 h. Uninoculated 100 cuttings and 100 scions were placed in an identical solution without bacterial cells. After inoculation, the bacterial concentration in both cuttings and scions was measured through plate CFU counts. Then, the plant materials were grafted. The endophytic presence of MP12 in the leaves was monitored both one (after the forcing period) and two months (after planting in the vine nursery field) from the inoculum by denaturing gradient gel electrophoresis (PCR-DGGE) technique. Eventually, the presence of *P. protegens* MP12 in roots was monitored at the plant uprooting by PCR using specific primer for *P. protegens* species, (ii) inoculum B: after the forcing period, 100 grafted rooted plants were placed in MP12 suspension for seven days. The control samples were immersed in an identical solution without bacterial cells for the same period of time. Afterwards, the bacterial concentration in the vine plants was analyzed by plate CFU counts. Hence, plant materials were planted in a vine nursery field, and the presence of MP12 strain was assessed both in leaves and roots as above-mentioned, after approximately one month and at the end of the experiment, and lasted about seven months after the inoculum.

#### 2.4.2. Rhizospheric Inoculation Treatments

The rhizospheric treatment was carried out after one month from the planting of the grafted plants in the vine nursery field: 50 mL of *P. protegens* MP12 cell suspension (obtained as above-mentioned) were injected in the soil close to the roots of 100 vine plants. The presence of MP12 strain was monitored by PCR-DGGE immediately after the inoculum and one month later. Eventually, the persistence of the inoculated strain was assessed at the plant uprooting (six months after inoculation) through PCR using specific primers for *phlD* gene [24].

#### 2.4.3. Epiphytic Inoculation Treatments

The epiphytic inoculation using dsRed-tagged MP12 strain was carried out and monitored in a greenhouse in order to avoid weather variables such as rain and storm. The strain was grown for 48 h at 27 °C (200 r.p.m.) and collected by centrifugation (4500× *g* for 20 min at 4 °C) and cells were resuspended in sterilized water (about 10^8^ CFU mL^−1^). The bacterial inoculum was thus sprayed on the leaves of five young plants (more than 50 leaves in total). The control treatment was performed on the same number of plants by using sterilized water. The persistence of MP12 strain was monitored after 24, 72, and 168 h after the inoculum.

### 2.5. Preparation of Plant Samples and Bacterial Counts

#### 2.5.1. Post Endophytic Inoculation Treatments

Ten plants for each treatment were individually sampled and analyzed. The presence of MP12 after both inoculum A and B was assessed in the first internode of the stem. The plant materials were surface-sterilized for one min with ethanol 70% and then exposed for three min to NaOCl (3% [*w*/*v*] chlorine). Afterwards, the stems were rinsed three times for five min with sterile physiological solution (0.9% [*w*/*v*] NaCl). Then the tissues were finely cut with a sterile scalpel and collected in 10 mL sterile tubes supplied with five mL of physiologic solution. The tubes were vortexed for one min, placed for one h on an orbital shaker (250 r.p.m.), and then vortexed again for one min. The bacterial counts were performed through plating serial dilutions on King B agar medium [19], and the CFUs g^−1^ was calculated. Eventually, the disinfection protocol was verified by plating 100 µL of physiological solution from the third rinse on agarized King B medium.

#### 2.5.2. Post Epiphytic Inoculation Treatments

Ten leaves were randomly harvested from both control and dsRed-tagged MP12 inoculated plants and placed in 10 mL sterile tubes. Thereafter, five mL of physiological solution were added to each tube, vortexed for one min, placed for one h on an orbital shaker (250 r.p.m.), and then vortexed again for one min. Serial dilutions were plated on King B agar medium; dsRed-tagged MP12 colonies were observed and counted through Leica MZ16 F fluorescent stereomicroscope.

### 2.6. Recovery of Inoculated Bacteria through Molecular Analysis

The monitoring of *P. protegens* MP12 after both endophytic and rhizospheric treatments was carried out even by culture-independent techniques: denaturing gradient gel electrophoresis (PCR-DGGE) and PCR by using a *P. protegens* specie-specific primer pair for *phlD* gene responsible for the synthesis of the antifungal compounds 2,4-diacetylphloroglucinol (2,4-DAPG) [24].

The recovery of MP12 strain after endophytic treatment was performed after the forcing period and after one month of permanence in the vine nursery field. Leaves from 10 plants were collected, cleaned, and surface sterilized as above described. The permanence of MP12 strain after rhizospheric treatments was ascertained four hours and one month from the inoculum. The samples were analyzed through a quartering procedure [25]. Total DNA was thus extracted from the plant materials and soil by using FastDNA SPIN for Soil Kit (MP, Biomedicals) [26]. The PCR-DGGE was carried out as described in Andreolli et al. [18].

The uprooted vine plants were analyzed in order to verify the presence of *P. protegens* MP12 in the rhizosphere and the roots from samples treated with the bacterium. Surface-sterilized roots of untreated plants were analyzed as control. Total DNA was recovered either from unsterilized roots with adhering soil particles or from sterilized roots by using FastDNA SPIN for Soil Kit as above-mentioned. The amplification of the specific *phlD* gene was performed as reported in Mavrodi et al. [24]. The amplicons were thus extracted from the agarose gel by using QIAEX II Gel Extraction kit, sequenced (GATC Biotech; Cologne, Germany) and searched for homology using BlastN against the NCBI database [27].

### 2.7. Evaluation of Grafted Plant Quality

The quality of the grafted plants was evaluated during the vegetative season by counting the number of dead plants and scoring the vegetative development (number and length of canes per plant). After the uprooting, the plants were ranked on the basis of the root development in two commercial groups (high- and low-grade).

### 2.8. Evaluation of Fungal Infection on Grapevine Leaves Treated with P. protegens MP12

An infection assay was carried out to evaluate the antagonist effect of *P. protegens* MP12 on artificially infected leaves of grapevine by *B. cinerea* BC. Leaves of young grapevine plants treated with the bacterium by spraying a cell suspension of dsRed-tagged MP12 and with sterilized water, as above described, were used for the assay. A total of 60 detached leaves (30 from bacterial treated plants and 30 from control plants) were infected using a *B. cinerea* BC plugs from a culture of seven days. Three plugs were separately placed mycelium-side down onto the leaf surface to have three replicates on each leaf, which was placed in a sterile petri dish containing water soaked sterilized paper and incubated at 25 °C. The disease severity was scored on a scale of 0–6 based on necrotic area developed around the plug after two weeks of incubation, and disease index (DI) was calculated [28].

### 2.9. Copper Sensitivity Assay

A cell suspension of *P. protegens* MP12, grown for 48 h at 27 °C (200 r.p.m.), was transferred reaching a final OD_600_ of 0.01 to 5 mL of King B medium spiked with 50, 100, and 150 mg L^−1^ of Cu^2+^ supplied as CuSO_4_∙ 5 H_2_O. The same media without inoculum was used as negative control and reference for the determination of MP12 growth by the measure of OD_600_.

### 2.10. Statistical Analysis

The *t*-test, performed using the statistical package XLSTAT 2017 (Addinsoft SARL, Paris, France), was used to determine significance (*p* < 0.05) between untreated (control) and treated samples with bacterial suspension about the cell enumeration, quality evaluation of grafted plants, and disease index on leaves infected by *B. cinerea*.

## 3. Results

### 3.1. Construction and Analysis of dsRed-Tagged P. protegens MP12 Strain

The colony PCR screening was performed with glmSdown and Tn7R primers, while further sequencing analysis confirmed the correct site-specific gene insertion (data not shown). Colonies and cells of the dsRed-tagged *P. protegens* MP12 strain displayed vivid red fluorescence (Figure 1). No wild-type colonies appeared after several purification on King B agar medium without the appropriate antibiotic, indicating a stable integration of the mini-transposon into the bacterial chromosome. It was observed that the PGPR properties were fully maintained in dsRed-tagged *P. protegens* MP12.

### 3.2. Endophytic Colonization

#### 3.2.1. Transformed dsRed-Tagged *P. protegens* MP12 in Root Tissue

Rooted grapevine cuttings were inoculated with dsRed-tagged *P. protegens* MP12 in order to verify the ability of this strain to colonize the plants through the roots (Figure 2). The results confirmed fluorescent cells into the root tissue of cuttings placed for one week in a suspension of dsRed-tagged *P. protegens* MP12. No fluorescent microorganisms were observed in roots that had not been inoculated.

#### 3.2.2. Colonization of Vine Plants by *P. protegens* MP12

Both microbiological and molecular analysis were conducted on samples collected at random from each group of vine plants inoculated with *P. protegens* MP12 (inoculum A, inoculum B, rhizospheric) in order to check the potential ability to colonize the inner tissues of vine plants.

In inoculation A, unrooted cuttings and scions were dealt with separately. Plate counts performed 48 h after treatment revealed a significant difference (*p* < 0.01) between the inoculated cutting samples (4.75 ± 0.72 CFUs cm^−1^) and the control specimens (2.66 ± 0.23 CFUs cm^−1^). However, no significant differences (*p* > 0.05) in bacterial counts were observed between the inoculated (3.01 ± 1.47 CFUs cm^−1^) and control (4.26 ± 0.10 CFUs cm^−1^) scions (Figure 3).

After the grafting and a forcing period of one month, DGGE analysis with band sequencing revealed evidence of potential *P. protegens* MP12 only in young leaves collected from the treated samples but not in untreated plants (Figure 4A). The grafted vine plants were then planted in a nursery field, and DGGE analysis of the leaves was conducted one month later, when the MP12 strain was no longer detectable (Figure 4B). Similarly, no amplicons were obtained from the specific PCR performed on *phlD* gene in uprooted plants in leaves, rhizosphere and roots of the inoculated plants (data not shown). The results therefore suggest that *P. protegens* MP12 was able to colonize the cuttings and the entire plant until the end of the forcing period, but once the plant was in the nursery field, its persistence was strongly affected.

In the inoculation protocol B, the MP12 strain was inoculated in vine plants after the forcing period. Plate counts conducted seven days after inoculum displayed a cell concentration of 4.83 ± 0.63 and 3.38 ± 0.19 CFUs cm^−1^ (*p* < 0.05) in MP12 inoculated and control plants, respectively (Figure 3). As with inoculum A, *P. protegens* MP12 was no longer detected, either in leaves, using PCR-DGGE after one month of growth in the vine nursery field (Figure 4B), or in the rhizosphere or roots, on the application of specific PCR after the uprooting (data not shown).

The MP12 strain may therefore be adsorbed by the vine plants, but, as with plants treated with inoculation protocol A, *P. protegens* MP12 populations settled into the inner tissues disappeared from plants in the nursery field.

### 3.3. Rhizospheric Inoculum

For the rhizospheric inoculation, rooted vine plants were treated in the soil 30 days after planting in the nursery field. Due to the high number of autochthonous bacteria in the soil, a plate count was not performed after inoculum. In order to verify the permanence of the MP12 strain in the rhizosphere, PCR-DGGE analysis was carried out four hours and one month after inoculum. The results clearly showed a band referring to a potential MP12 strain four hours after inoculum. Moreover, a reduced but still detectable band was observed in all treated samples 30 days after treatment (Figure 5). The presence of MP12 in the vine plant rhizosphere was confirmed when the plant were uprooted at the end of vegetative season. A single amplicon derived from PCR performed on *phlD* gene was obtained from unsterilized roots with adhering soil particles. Sequence analysis confirmed the exact correspondence with the deposited homologous gene of *P. protegens* MP12 (Acc. Number KX236067) (data not shown). However, no amplification was observed from DNA samples extracted from surface-sterilized roots, suggesting the absence of the strain in the rhizosphere and on the surface of the roots.

### 3.4. Quality Evaluation of Grafted Plants

An evaluation was carried out on grafted plants treated with rhizospheric inoculum, since the MP12 persistence at uprooting was assessed only in these plants. No significant differences were observed between treated grafted plants and control plants, based on the number of dead plants and the score of vegetative development (data not shown).

### 3.5. Epiphytic Inoculum, Fungal Infection Assay, and Copper Sensitivity Test

In addition to the inoculation of *P. protegens* MP12 on vine plants, the dsRed-tagged MP12 strain trials was sprayed on leaves of young grapevine plants. Plate counts carried out after one, three, seven, and 14 days from treatment revealed concentrations of 7.84 ± 0.18, 5.85 ± 0.53, 5.04 ± 0.39 and 3.88 ± 1.83 log(CFU g^−1^) of MP12, respectively (Figure 6).

The infection caused by *B. cinerea* BC plugs on the lower surface of the detached leaves of plants treated with MP12 three days after bacterial inoculum was significantly (*t*-test, *p* < 0.001) lower than in the control plants (DI was 48.7 and 29.8%, respectively).

The growth curve of *P. protegens* MP12 in presence of copper showed that the bacterium was only slightly inhibited at level up to 100 mg L^−1^ of Cu^2+^, but completely suppressed at 150 mg L^−1^ (Figure 7).

## 4. Discussion

In this study, different inoculation protocols of *P. protegens* MP12 were adopted to evaluate its ability to colonize vine plants under unsterile conditions. Experimental conditions were similar to those in a standard process of grapevine nursery propagation. Endophytic colonization showed that the *P. protegens* MP12 remained in vine plants until the end of the forcing period. However, the bacterium did not last long and the cell population disappeared after vine plants were planted in the nursery field.

The ability of *P. protegens* MP12 to efficiently penetrate and colonize the inner tissues of grapevine had already been observed in vitro using micropropagated plantlets under sterile conditions [18]. The presence of dsRed-tagged *P. protegens* MP12 cells in the inner part of the radicle confirmed its ability to penetrate inside the tissue. Although not clearly defined under confocal microscope, dsRed-tagged MP12 could be seen in the cortex and near the endodermis barriers one week after inoculation, as reported by Compant et al. [17] and Isaur-Kruh et al. [29] using *B. phytofirmans* PsJN and *Dyella*-like strains, respectively. *Pseudomonas* is a genus with a known ability to colonize the tissue of grapevine or other plants such as poplar trees [30,31].

The results here obtained showed that MP12 strain was able to spread to young leaves during a forcing period of about one month. Endophytic colonization of aerial plant parts was found to require more time compared with the in vitro system [17]. This can be due to the different plant physiology between the two systems: a primarly xylem only occurs in plants grown in vitro compared to vine cuttings [32]. However, it was shown that bacterial can be disseminated through the xylem structures of grapevines through passive mechanisms [32,33]. It has already been demonstrated that inoculated bacteria in vine cuttings appeared in the root tissues after 48-72 h after inoculation, reaching the upper part of the plant (grape inflorescence stalks and in young berries) after five to six weeks [31]. Isaur-Kruh et al. [29] reported that *Dyella*-like bacterium can reach shoots and leaves through several internodes one week after the inoculation of vine cuttings. This bacterium can penetrate different plant organs but cannot survive for more than six weeks [29].

Once the plant materials were placed in the vine nursery field, the MP12 strain was no longer detectable. According to Álvarez-Pérez et al. [34], the degree of endophytic colonization of actinobacteria in vine plants tends to decrease over time. It is probable that endophytic indigenous microflora drastically increases microbiological competition in the colonization of plant tissues and negatively influences the settlement of exogenous bacteria, like *P. protegens* MP12 in plant tissues [5,31]. The ability of allochthonous strains to overcome competition with indigenous microorganisms in the roots, xylem, and leaves is fundamental for a stable endophytic colonization [5]. It is probable that the rhizospheric origin of *P. protegens* MP12 and so its physiologic adaptation for this habitat may impede its survival in plans tissues, where better adapted microflora can be found at concentrations ranging from about 1.5 to 6 CFU g^−1^ [5]. Moreover, the ability of endophytic bacteria to avoid stimulating the plant’s defensive response is fundamental for their colonization in plant tissue [35,36]. Specific receptors on the surface of plant cells can recognize conserved bacterial molecules (e.g., flagellin and lipopolysaccharides), triggering a defence response by the plant [37,38]. However, endophytes can evade recognition by plants or induce only a weak and transient defensive reaction compared to pathogenic infection [39]. In the event of the generation of reactive oxygen species (ROS) as a plant defensive system, endophytes can protect themselves by producing enzymes, such as catalases and peroxidases [40]. For instance, a large amount of ROS-deactivating genes was expressed in *Gluconacetobacter diazotrophicus* during the initial stages of colonization [41]. It was also shown that gumD gene, which is involved in exopolysaccharide production, was required for biofilm formation and efficient plant colonization. The ability of the MP12 strain to colonize grapevines grown in vitro and vine cuttings suggests that this strain may be able to completely partially avoid a plant’s defensive system. On the other hand, the presence of potentially pathogenic bacterial strains in the vine nursery soil [42] may trigger a plant’s defensive system and mechanisms [36], thereby also affecting the persistence of the MP12 strain in plants.

Analysis conducted when the plants were uprooted showed that the *P. protegens* MP12 strain is able to colonize the rhizosphere. This property appears to be consistent with the origin of *P. protegens* MP12 [18] and the previous identification of other root-colonizing strains from the same species [42]. This strain’s ability to colonize the rhizoplane underlines the importance of the isolation source in the selection of efficient and beneficial plant bacteria. Elevated rhizosphere competence in an inoculant bacterial strain has been shown to be a key factor for the improvement of plant health and the suppression of phytopathogens [14,43,44]. In particular, the presence of *P. protegens* MP12 in the rhizosphere may be of particular importance, since it has been shown that most of vine plants can be infected through the roots, once placed in an open-field nursery [45]. Halleen et al. [45] observed that 1% or less of plants were infected with *Cylindrocarpon* spp.—the agent that causes young grapevine decline (YGD)—in the nursery greenhouse, whereas more than 50% were infected at the end of the season. On the other hand, further investigations are required to evaluate the beneficial phytoprotection effects of MP12 strain in the rhizosphere of young grapevines.

Foliar application trials demonstrated that *P. protegens* MP12 can be a suitable biocontrol agent in epiphytic treatments. Leaf spraying of bacteria is a well-known biological alternative to treatment with chemical pesticides against plant diseases [46]. In this study, leaf spraying of *P. protegens* MP12 significantly reduced fungal infection caused by *B. cinerea* under unsterile conditions that simulated foliar spraying in the field. Different *Pseudomonas* species, especially *P. fluorescens*, which is closely related taxonomically to *P. protegens* [47,48], display biocontrol activity when inoculated by foliar spraying [46,49,50]. The low infection of *B. cinerea* in treated grapevine plants may be due to the high amount of *P. protegens* MP12 detected on the leaves in microbiological analysis conducted after foliar spraying. Fu et al. [51] found about 2.5 log(CFU g^−1^) of *Bacillus* sp. strains after 14 days after foliar spraying, a very low concentration when compared with the data reported here. It is not known whether this bacterium can penetrate the leaf cuticle so further analyses are required. The endophytic colonization by microorganisms of the leaf via stomata has barely been investigated, but there is evidence that some bacteria, including plant growth-promoting strains, are able to enter the stomata as endophytes [52].

An in vitro experiment was conducted in order to evaluate the compatibility of copper-based fungicides—commonly used in biological/integrated grapevine management—with *P. protegens* MP12 biocontrol agents. In order to plan an effective disease management strategy, potential bioagents must be compatible with fungicides. Combinations of biocontrol agents and fungicides reduce the need for chemical applications and may be a practicable method to control *B. cinerea* [53] and Fusarium graminearum [54], for example. According to European Council Directive 86/278/EEC [55], the limit values of this metal in soil range from 50 to 140 mg kg^−1^ of dry matter. Since the repeated use of copper-based fungicides leads to the accumulation of copper in the soil, *P. protegens* MP12, based on the result of copper sensitivity test, may be entirely compatible in biological/integrated grapevine management where copper-based fungicides are commonly used.

## 5. Conclusions

The applicative approach of this study demonstrated the success of rhizospheric colonization of *P. protegens* MP12 in vine plants through soil inoculation when these vines were planted in the nursery. However, the plant defence mechanisms and competition from indigenous microorganisms are probable reasons for the failure of permanent endophytic colonization induced by inoculation of rootstocks, scions, and grafted rooted plants. Further investigations are required to check whether the rhizospheric colonization of the MP12 strain is stable in vines once planted in vineyards. Moreover, foliar application by spraying *P. protegens* MP12 bacterial suspension would appear to be a valid alternative biocontrol treatment against fungal pathogens in field, although further trials are recommended in order to confirm its effectiveness in vineyards. Finally, further investigations are needed to evaluate the physiologic interaction between MP12 strain and plants and to analyze the beneficial phytoprotection effects of this strain in the rhizosphere of young grapevine.

## Figures and Tables

**Figure 1 microorganisms-09-00234-f001:**
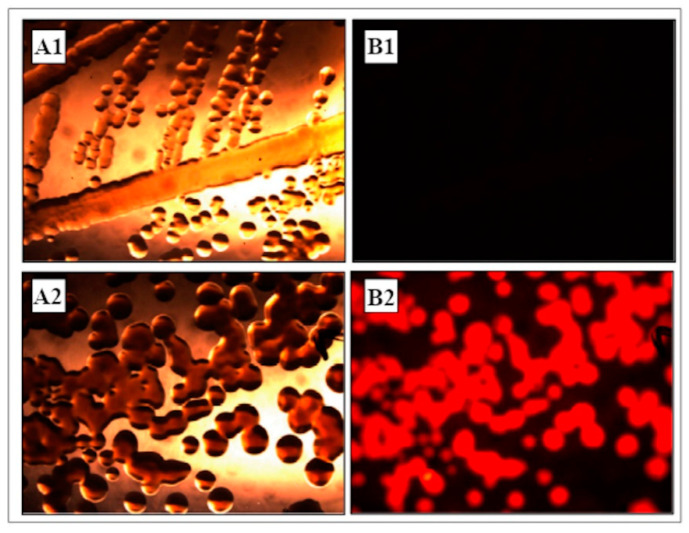
(**A**) Colonies of *Pseudomonas protegens* MP12 and (**B**) dsRed-tagged MP12 strain observed under a stereo microscopy by using (1) white light and (2) fluorescent light.

**Figure 2 microorganisms-09-00234-f002:**
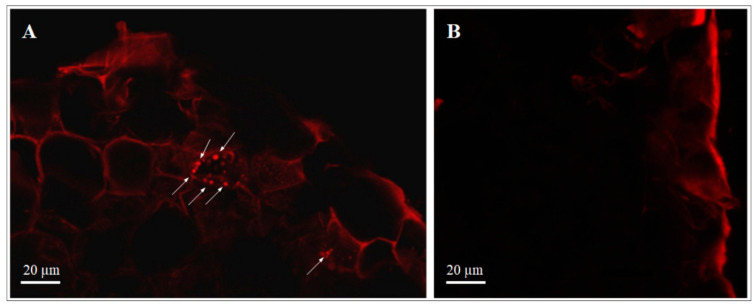
Images under the confocal microscope of roots of rooted grapevine cuttings inoculated with (**A**) dsRed-tagged *P. protegens* MP12 one week after inoculation. dsRed-tagged bacteria (arrows) were visualized in the cortex and near the endodermis barriers. (**B**) Fluorescent microorganisms were not observed in the control roots (uninoculated cuttings).

**Figure 3 microorganisms-09-00234-f003:**
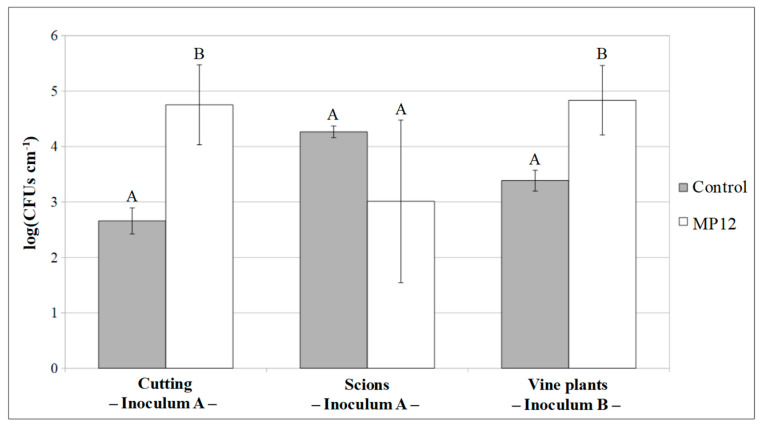
Values of the cell counts in the first internode of cuttings, scions (inoculum A), and vine plants (inoculum B) after endophytic inoculation. Mean and standard deviation of 10 plant samples were considered. Mean values marked by the same letter are not significantly different at *p* < 0.05.

**Figure 4 microorganisms-09-00234-f004:**
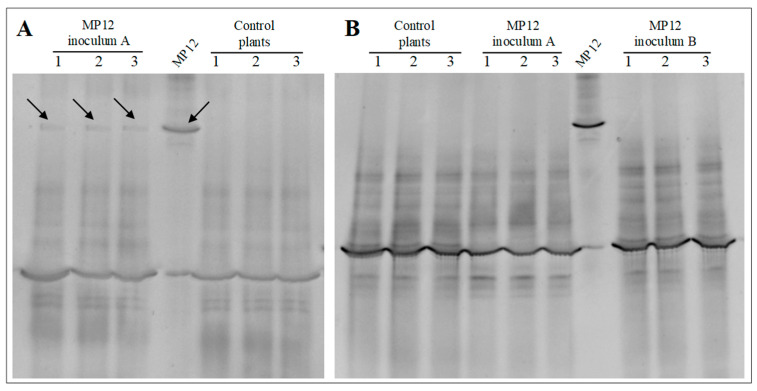
(**A**) PCR-DGGE analysis on leaves of both grafted vine plants inoculated with MP12 strain by endophytic inoculum A and uninoculated plants after the forcing period lasted one month. (**B**) DGGE analysis on leaves of plants inoculated with MP12 strain by endophytic inoculum A and B after one month of permanence in the vine nursery field. Fragment amplified from *P. protegens* MP12 was used as reference.

**Figure 5 microorganisms-09-00234-f005:**
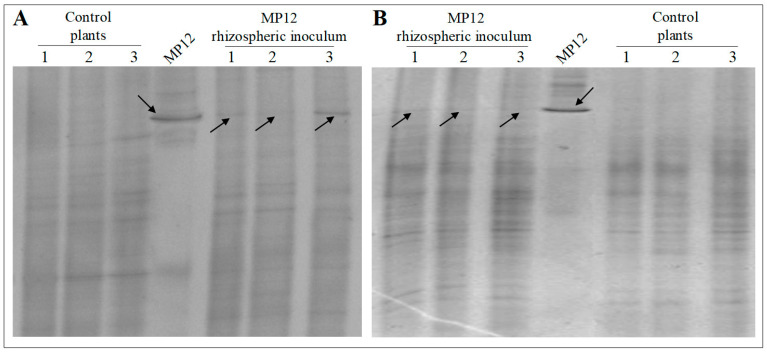
PCR-DGGE analysis of the rhizosphere of grapevine plants placed in the nursery field after (**A**) four hours and (**B**) one month after the MP12 rhizospheric inoculum. DGGE analysis of uninoculated plants (control plants) are also reported. Fragment amplified from *P. protegens* MP12 was used as reference.

**Figure 6 microorganisms-09-00234-f006:**
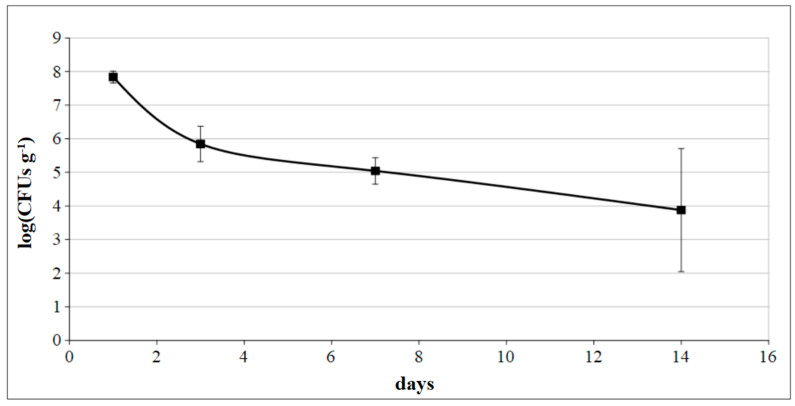
Leaves surface colonization of dsRed-tagged *P. protegens* MP12 strain after epiphytic inoculation performed by foliar spray.

**Figure 7 microorganisms-09-00234-f007:**
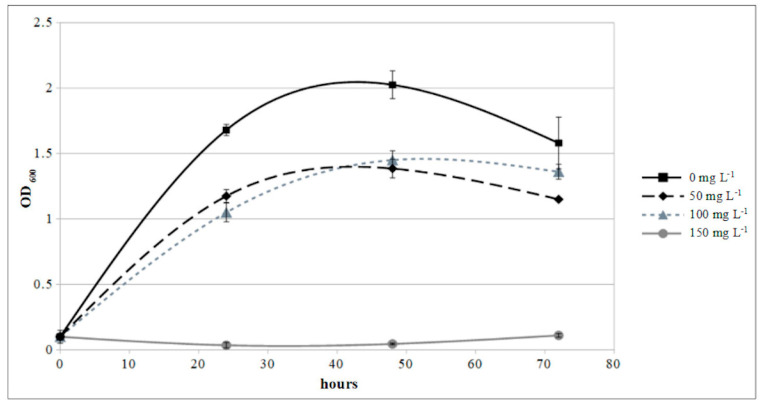
Time courses of microbial growth of *P. protegens* MP12 in presence of 0, 50, 100, and 150 mg L^−1^ of Cu^2+^.

## Data Availability

Data is contained within the article. Nucleotic sequences of *Pseudomonas protegens* MP12 are openly available in NCBI at reference number KJ467788 for 16*S* rRNA gene and KX236067 for *phl*D sequence.

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
