# Peer review of "In Vivo Endophytic, Rhizospheric and Epiphytic Colonization of Vitis vinifera by the Plant-Growth Promoting and Antifungal Strain Pseudomonas protegens MP12"

_microorganisms, 2021, doi:10.3390/microorganisms9020234_

Round 1

Reviewer 1 Report

Grapevines are a popular fruit crop all over the world. Grape vine diseases caused by phytopathogens can have serious negative impact on vineyard’s life and productivity. Due to increasing concern about the use of chemical pesticides and pesticide residues, sustainable alternative approaches such as the use of biological control agents is gaining more acceptance.

The manuscript titled – In vivo endophytic, rhizospheric and epiphytic colonization of Vitis vinifera by the plant-growth promoting and antifungal strain Pseudomonas protegens MP12 by Marco Andreolli et al. describes an evaluation of three different inoculation methods on various colonization pattern of MP12. Colonization of grape plant by P. protegens MP12 was monitored in multiple ways (fluorescent marker protein dsRed, fluorescent stereomicroscopy, confocal microscopy, PCR using specific primer for P. protegens species, and PCR-DGGE analysis). The assessment of various inoculation methods was conducted under non-sterile conditions, which are more realistic environmental conditions, thereby increasing the ecological validity of the experimental results.

Antagonism against fungal pathogen Botrytis cinerea using a detached whole leaf bioassay and copper sensitivity assay studies were also conducted.

Major concerns:

  1. Pseudomonas protegens MP12 was originally isolated from soil. Antagonistic microorganisms (endophytes) already live in the plant tissue without causing damage to the plant. An endophyte already adopted to its host plant (grape) and its agroecosystem would be an ideal candidate. The authors themselves have emphasized the importance of this aspect in the manuscript (Line 51-52).

  1. To provide a clear understanding (especially the “Materials and Methods” section), please carefully proof-read and eliminate issues related to spelling, grammar, phrasing, and proper technical terms. Use “uninoculated” instead of “Not-inoculated.” Similarly, use “cell-free culture filtrate” instead of “free bacterial cell suspension.” Better to use “colonize” instead of “infect (Line 368).”

  1. Line 114: “roots of ten young plants” How old were these plants? Were the roots of those plants free of adhering soil particles?

  1. Line 123 – 124: “ten plant or soil samples” “plant tissues” from where did this plant and soil samples come from? Plant sample or plant tissue – were those root samples/tissue or shoot samples/tissue? Need more details.

  1. Line 131-132: Please clarify this statement – “rhizosphere and no sterilized roots were sampled together – or in the surface-sterilized roots.”

  1. Need a section on statistical analysis.

Author Response

We thank the reviewer for her/his comments useful to improve the manuscript.

Pseudomonas protegens MP12 was originally isolated from soil. Antagonistic microorganisms (endophytes) already live in the plant tissue without causing damage to the plant. An endophyte already adopted to its host plant (grape) and its agroecosystem would be an ideal candidate. The authors themselves have emphasized the importance of this aspect in the manuscript (Line 51-52).

Authors: This aspect was included in the introduction section. A sentence was added (L60-63).

To provide a clear understanding (especially the “Materials and Methods” section), please carefully proof-read and eliminate issues related to spelling, grammar, phrasing, and proper technical terms. Use “uninoculated” instead of “Not-inoculated.” Similarly, use “cell-free culture filtrate” instead of “free bacterial cell suspension.” Better to use “colonize” instead of “infect (Line 368).”

Authors: The spelling, grammar and technical terms were checked and modified according to reviewer’s suggestions. Moreover, an accurate review of the English language was made.

Line 114: “roots of ten young plants” How old were these plants? Were the roots of those plants free of adhering soil particles?

Authors: The age and details on roots were included (L130-131).

Line 123 – 124: “ten plant or soil samples” “plant tissues” from where did this plant and soil samples come from? Plant sample or plant tissue – were those root samples/tissue or shoot samples/tissue? Need more details.

Authors: The entire paragraph was improved by adding more details (L214-233).

Line 131-132: Please clarify this statement – “rhizosphere and no sterilized roots were sampled together – or in the surface-sterilized roots.”

Authors: the sentence was modified in order to be clear (L226-228).

Need a section on statistical analysis.

Authors: a section with the statistical analysis was included (L259-263).

Reviewer 2 Report

The article is well written in an appropriate way with the data presented in appropriate form. This is an important and an active area of research and a better understanding of the role of PGPR for plant growth under different inoculations and environments. This topic is one that should be of general interest to readers of Microorganisms, though I do have concerns with manuscript that I detail below.

1, The comparison of the endophytic, rhizospheric and epiphytic colonization was not under the same experiment using consistent methods. So there is no statistical support to argue the difference between the three colonizations, although certain discussions are still OK.

2, There are thousands studies on testing plant beneficial microbes, we all know that most were unsuccessful when transporting from agar to the soils. So I expect more discussions on the mechanisms for the poor colonisation/performance of PGPR in the field.

3, PCR-DGGE method might be able to identify the bacteria at the species level, but not the strain level. Please see the detail methodology discussed from doi.org/10.1016/j.soilbio.2018.12.012 So I think the identified strain here can be called “potential” P. protegens MP12.

4, The dsRed-tagged P. protegens MP12 should have the same PGPR function as the wild, so related genomic evidence or data on plate performance should be provided.

Author Response

We thank the reviewer for her/his comments useful to improve the manuscript.

The comparison of the endophytic, rhizospheric and epiphytic colonization was not under the same experiment using consistent methods. So there is no statistical support to argue the difference between the three colonizations, although certain discussions are still OK.

Authors: The reviewer has right. We have used different methods to induce the colonization in plants based on the three types of bacterial treatment (endophytic, rhizospheric and epiphytic). This experimental design was chosen as the most suitable technique for each treatment. Data of this work are useful for further investigations where same experimental conditions for each type of colonization could be applied. Statistical analysis was carried out for enumeration (plate count) in endophytic (L298-303; L345-347) and epiphytic (L397-399) colonization, while rhizopheric colonization was assessed by qualitative analysis (PCR-DGGE, PCR with species-specific primers). As above said, because the experimental conditions were not similar for all type of colonization protocols, statistical analysis to argue difference among them was not possible.

There are thousands studies on testing plant beneficial microbes, we all know that most were unsuccessful when transporting from agar to the soils. So I expect more discussions on the mechanisms for the poor colonisation/performance of PGPR in the field.

Authors: More discussion regarding this topic was included in the text (L451-455; L468-495).

PCR-DGGE method might be able to identify the bacteria at the species level, but not the strain level. Please see the detail methodology discussed from doi.org/10.1016/j.soilbio.2018.12.012 So I think the identified strain here can be called “potential” P. protegens MP12.

Authors: The reviewer has right. “Potential” P. protegens MP12 was included in the text (L318; L363).

The dsRed-tagged P. protegens MP12 should have the same PGPR function as the wild, so related genomic evidence or data on plate performance should be provided.

Authors: The mini-Tn7 system was here chosen because chromosomal single-copy and site-specific insertion allows to perform experiments in environments where antibiotic selection is not feasible without compromising the host cell fitness and performance (L92-95). However, although in the previous version of the manuscript was not included, PGPR properties of the transformed strain were tested. These details were here described both in materials and methods and results, where a specific paragraph was included (L111-112; L267-268).